# QTL Mapping for Wheat Seed Dormancy in a Yangmai16/Zhongmai895 Double Haploid Population

**DOI:** 10.3390/plants12040759

**Published:** 2023-02-08

**Authors:** Gang Guo, Shuhao Xu, Hao Chen, Yuanfeng Hao, Hailiang Mao

**Affiliations:** 1National Key Laboratory of Crop Genetic Improvement, Huazhong Agricultural University, Wuhan 430070, China; 2College of Plant Science and Technology, Huazhong Agricultural University, Wuhan 430070, China; 3Institute of Crop Sciences, Chinese Academy of Agricultural Sciences (CAAS), 12 Zhongguancun South Street, Beijing 100081, China

**Keywords:** wheat, seed dormancy, pre-harvest sprouting, QTL

## Abstract

Pre-harvest sprouting (PHS) of wheat reduces grain yield and quality, and it is strongly affected by seed dormancy. Therefore, identification of quantitative trait loci (QTL) for seed dormancy is essential for PHS resistance breeding. A doubled haploid (DH) population, consisting of 174 lines from the cross between Yangmai16 (YM16) and Zhongmai895 (ZM895) was used to detect QTLs for seed dormancy and grain color. For seed dormancy, a total of seven QTLs were detected on chromosomes 2A, 3A, 3D, 4D, 5B and 5D over four environments, among which *Qdor.hzau-3A*, *Qdor.hzau-3D.1* and *Qdor.hzau-3D.2* were stably detected in more than two environments. For grain color, only two QTLs, *Qgc.hzau-3A* and *Qgc.hzau-3D* were detected on chromosomes 3A and 3D, which physically overlapped with *Qdor.hzau-3A* and *Qdor.hzau-3D.1*, respectively. *Qdor.hzau-3D.2* has never been reported elsewhere and is probably a novel locus with allelic effect of seed dormancy contributed by weakly dormant parent ZM895, and a KASP marker was developed and validated in a wheat natural population. This study provides new information on the genetic dissection of seed dormancy, which may aid in further improvement for marker-assisted wheat breeding for PHS resistance.

## 1. Introduction

Bread wheat (*Triticum aestivum* L.) is one of the most important cereal crops around the world and contributes nearly 20% of the total calories consumed by humans [1]. Pre-harvest sprouting (PHS), physiologically mature grains germination in spikes before harvest, causes huge loss in grain yield and quality, particularly in regions with harvest time overlapping with rainy season [2]. The economic yearly loss caused by PHS in wheat production is more than one billion dollars worldwide [3]. Thus, it is imperative to identify PHS related QTLs that can be used to breed for PHS-resistant wheat cultivars.

Many factors have been found to be associated with pre-harvest sprouting, with seed dormancy and grain color as two major factors [4,5]. Seed dormancy can be defined as an intact and viable seed that fails to germinate under favorable conditions [4]. In wheat, QTLs associated with seed dormancy have been found on all 21 chromosomes, most of which are enriched on chromosomes of the third and fourth chromosome homologous groups [6]. Several genes related to seed dormancy have been cloned through map-based cloning. *TaPHS1* (also called *TaMFT-3A*), the causal gene for the major QTL *Qphs.pseru-3AS*, was shown to be a positive regulator of ABA sensitivity and increases seed dormancy [7,8,9,10]. *TaMKK3* (*mitogen activated kinase kinase 3*), the causal gene for another major seed dormancy QTL *Phs1-4AL*, was involved in protein phosphorylation in signal transduction pathways and affected seed dormancy by positively modulating the responsiveness of ABA [11,12,13]. Moreover, several seed dormancy-related genes were cloned in wheat through homology-based cloning. *TaVp1*, a wheat ortholog of *Vp1* (*Viviparous-1*) gene in maize and *ABI3* (*ABA Insensitive 3*) in *Arabidopsis*, confers a high level of seed dormancy [14,15,16,17,18]. *TaSdr*, an ortholog of *OsSdr4* gene in rice, has a significant effect on seed dormancy with a missense mutation SNP643 which resulted in a substitution from Val to Ile [19,20]. *TaQsd1*, the ortholog of *Qsd1* in barley, which encodes an alanine aminotransferase, has been shown to be necessary in the early seed germination [21,22,23]. *TaDOG1*, the ortholog of *DOG1* (*Delay of Germination 1*) gene in *Arabidopsis*, was found to increase seed dormancy via ABA1-dependent inhibition of *Hypersensitive Germination 1* [24,25,26,27,28,29].

Grain color is another important factor for seed dormancy. Previous studies showed that red grain wheat is usually more resistant to PHS than white grain wheat [30,31,32]. The *R* genes (contains *R-A1*, *R-B1* and *R-D1*) affect grain color by regulating the expression of flavonoid biosynthesis genes, such as *CHS*, *CHI* and *F3H* [33]. *TaMyb10*, the casual grain color gene of the *R* loci, encodes a MYB type of transcription factor and has a large effect on both grain color and seed dormancy [32,34,35,36,37]. Notably, recent studies confirmed the function of *TaMyb10-3B* through CRISPR/Cas9 in the wheat variety “Fielder” [38]. Another grain color related gene, *TaDFR*, has different expression level between white grain and red grain wheat and has been found to be associated with PHS resistance [39,40].

In this study, we used a double haploid population derived from a strongly dormant cultivar Yangmai16 (YM16) with red grain color and weakly dormant cultivar Zhongmai895 (ZM895) with white grain color, to detect QTLs for grain color and seed dormancy by evaluating the germination index (GI) and germination percentage (GP). The genetic effect of the identified QTLs for seed dormancy was further confirmed in natural population and DH subpopulation according to grain color. A KASP marker was successfully developed based on the identified QTL *Qdor.hzau-3D.2* of seed dormancy and might be useful for PHS-resistance breeding.

## 2. Results

### 2.1. Phenotype Characterization and Analysis

GI and GP were assessed in the parental and DH lines over four environments. Positive significant correlation was observed among different environments for both GI and GP (Appendix A). An analysis of variance (ANOVA) results showed significant variance of GI and GP for both genotypes and environments, and the interaction between genotype and environment (Appendix A). The data showed that GI and GP values were distributed continuously and varied widely in DH population across the environments tested (Appendix A). The weakly dormant parent ZM895 showed higher GI and GP values across all environments, while the values for the strongly dormant parent YM16 varied among different environments with much lower GI and GP compared to that of ZM895 (Table 1 and Table 2). The average GI and GP of the DH lines was generally higher in LY21 than in other environments, which could be the result of over-matured seeds when harvesting. Broad-sense heritability was over 0.75 for each of the four environments for GI and GP. Taken together, the heritability for GI and GP across all the four environments was 0.88, indicating that these traits were mainly determined by genetic factors.

Grain color was shown to be a key factor affecting seed dormancy [32]. Seeds from environment LY21 and XY22 were assessed through the sodium hydroxide immersion method as white grains and red grains exhibited different color after soaking (Appendix A). In both environments, the white-grained lines had higher GI and GP than that of the red-grained lines. The Student’s *t*-test was employed to show that GI and GP was significantly different between red and white DH lines (Appendix A). These results confirmed and consisted with previous results that grain color is a major factor for seed dormancy.

### 2.2. QTL Mapping for Seed Dormancy and Grain Color

For GI and GP, a total of seven QTLs were detected on chromosomes 2A, 3A, 3D, 4D, 5B and 5D. Three stable QTLs on chromosomes 3A and 3D were reproducibly detected in more than two environments for both GI and GP, and they were named *Qdor.hzau-3A*, *Qdor.hzau-3D.1* and *Qdor.hzau-3D.2*, respectively (Table 3). According to the flanking markers and physical positions using Chinese Spring v1.0 genome as reference, *Qdor.hzau-3A* (709 Mb) and *Qdor.hzau-3D.1* (572 Mb) could be the loci reported previously, *TaMyb10-A* (703.91 Mb) and *TabMyb10-D* (570.80 Mb) [6], respectively. The locus *Qdor.hzau-3D.2* was not found in previous results and could be a novel QTL with the LOD value 3.35–7.36, explained by the phenotypic variation 4.98–10.34%. The QTL was detected at approximately 230 cM on chromosome 3D, with flanking markers *AX-110937331* and *AX-108883716*. Blast search results showed that *Qdor.hzau-3D.2* was located in the interval 593.62–600.15 Mb, which contained 96 high-confidence genes. According to tissue expression profiles from Wheat Omics 1.0 website (http://202.194.139.32/, accessed on 25 December 2022), two genes, *TraesCS3D02G514000* and *TraesCS3D02G514300*, were regarded as strong candidate genes for future research as they are mainly expressed in the grain. Interestingly, we found that the additive value of *Qdor.hzau-3D.2* was positive among all environments, hinting that the dormant allele was contributed by weakly-dormant parent ZM895. The Student’s *t*-test results showed a significant difference on GI and GP between lines with YM16 allele and ZM895 allele in this DH population except for WH22 environment (Figure 1).

Another four seed dormancy related QTLs were also detected in this study, and they were *Qdor.hzau-2A*, *Qdor.hzau-4D*, *Qdor.hzau-5B* and *Qdor.hzau-5D* (Table 3). However, each of the four QTLs could only be detected in one environment, indicating that they were not stable QTLs. Further experiments need to be done to confirm these QTLs.

For grain color QTL analysis, two QTLs on chromosomes 3A and 3D were detected in this population, and they were named *Qgc.hzau-3A* and *Qgc.hzau-3D*, explained by 19.25–21.11% and 30.25–30.44% phenotypic variation, respectively (Table 4). Based on the genetic position and physical interval of the two QTLs, *Qgc.hzau-3A* and *Qgc.hzau-3D* could be the same locus as *Qdor.hzau-3A* and *Qdor.hzau-3D.1*, covering the known grain color transcription factor *TaMyb10* on chromosomes 3A and 3D, respectively. We did not detect the grain color QTL on chromosome 3B probably because there was no polymorphism between YM16 and ZM895 (Appendix A).

### 2.3. Effects of Qdor.hzau-3D.2 on Seed Dormancy in DH Population

As the grain color has a major effect on seed dormancy, the effect of *Qdor.hzau-3D.2* on seed dormancy was assessed in this DH population when the grain color QTLs were present or absent. To do this, the DH population were divided into four genotypic classes. Class 1 includes the lines without either two grain color QTLs *Qdor.hzau-3A*/*Qgc.hzau-3A*, *Qdor.hzau-3D.1*/*Qgc.hzau-3D* or *Qdor.hzau-3D.2*, while class 2 includes the lines without the two grain color QTLs but with *Qdor.hzau-3D.2*. Class 3 includes the lines with two grain color QTLs but without *Qdor.hzau-3D.2*, while class 4 includes the lines with both the two grain color QTLs and *Qdor.hzau-3D.2* (Table 5).

Based on the comparison results of four different classes, the GI and GP value of class 1 was higher than that of class 2, and the GI and GP value of class 3 was higher than that of class 4 (Table 5 and Figure 2A–D). The Student’s *t*-test showed a significant difference between class 1 and class 2 except for WH22 (Figure 2A,B), while significant difference was detected between class 3 and class 4 only in WH22 (Figure 2C,D). These results showed that the effect of *Qdor.hzau-3D.2* had a minor but stable effect on seed dormancy, and that the effect would be expressed only in the absence of *Qdor.hzau-3A*/*Qgc.hzau-3A* and *Qdor.hzau-3D.1*/*Qgc.hzau-3D*, while it would be suppressed when the grain color related QTLs were present.

In order to further investigate whether the effect of *Qdor.hzau-3D.2* is related to grain color, the effect for *Qdor.hzau-3D.2* in red and white subpopulations was analyzed based on the grain color assessment results in LY21 and XY22. We found that lines with two different alleles of *Qdor.hzau-3D.2* showed significantly different seed dormancy in both red and white grain subpopulations, and lines with allele from ZM895 showed stronger seed dormancy than that with allele from YM16 (Figure 3). These results confirmed that the effect of *Qdor.hzau-3D.2* on seed dormancy is independent of grain color, although it could be suppressed when the grain color related QTLs were present as aforementioned.

### 2.4. QTL Validation of Qdor.hzau-3D.2

The QTL *Qdor.hzau-3D.2* was identified as a novel QTL in this study and was stably detected; therefore, the flanking marker *AX-108883716* was converted to a KASP marker. This marker was used to validate the effect of *Qdor.hzau-3D.2* in a wheat natural population, containing 305 wheat cultivars and 107 wheat landraces, and spike germination percentage (SGP) [1] for this population was measured previously. This marker successfully classified the population into two groups, either with T allele from the ZM895, or C allele from the YM16 of *Qdor.hzau-3D.2* (Appendix A). The frequency of T and C allele in this population were nearly 60.7% and 39.3%, respectively. Overall, the lines with T allele showed lower SGP values than that with the C allele, although no significant difference was detected either in all wheat varieties or in subpopulations of cultivars, the significant difference was detected in subpopulation of landraces (Figure 4).

## 3. Discussion

Pre-harvest sprouting (PHS) is a complex quantitative trait that is affected by many factors, while seed dormancy and grain color are two important factors. PHS has caused huge loss of yield and quality in wheat production, so it is imperative to explore the QTLs related to seed dormancy for breeding PHS resistant wheat varieties and around 200 QTLs have been identified [6]. In this study, three stable QTLs were detected on chromosomes 3A and 3D: *Qdor.hzau-3A*, *Qdor.hzau-3D.1* and *Qdor.hzau-3D.2*. Comparative analysis showed that *Qdor.hzau-3A* and *Qdor.hzau-3D.1* were colocalized with grain color genes *TaMyb10-A* and *TaMyb10-D*, respectively. Notably, based on grain color mapping results, the QTLs *Qdor.hzau-3A* and *Qdor.hzau-3D.1* were co-localized with *Qgc.hzau-3A* and *Qgc.hzau-3D*, respectively. A significant difference on GI and GP was found between white and red DH lines, indicating grain color has a large effect on seed dormancy. These results confirmed the effect of grain color loci as reported in the previous studies [34,36,37].

The QTL *Qdor.hzau-3D.2* was identified as a novel stable seed dormancy related QTL in this study, with a dormant allele from weakly dormant parent ZM895. *Qdor.hzau-3D.2* improved seed dormancy only when the grain color-related QTLs *Qdor.hzau-3A/Qgc.hzau-3A* and *Qdor.hzau-3D.1/Qgc.hzau-3D* were absent, suggesting that the effect of *Qdor.hzau-3D.2* is stable but minor and that there might be an epistatic effect between them. In order to dissect the genetic interaction between *Qdor.hzau-3D.2* and *Qdor.hzau-3D.1/Qgc.hzau-3D* near isogenic lines (NIL) containing either one or both of them, will be developed and further investigated. It will also be used for fine-mapping the causal gene underlying the QTL *Qdor.hzau-3D.2*. In addition, *Qdor.hzau-3D.2* showed a significant difference on GI and GP values in both the red subpopulation and white subpopulation of this DH population, suggesting this QTL might affect seed dormancy independent of grain color. A KASP marker was successfully developed based on the flanking marker *AX-108883716* at *Qdor.hzau-3D.2*, and the marker successfully separated a wheat natural population into two groups. Based on the Student’s *t*-test results in the wheat natural population, it was shown that this marker was tightly associated with spike germination rate in the wheat landraces while not in cultivars, which further confirmed the minor effect on seed dormancy. Nonetheless, this KASP marker might be useful for marker-assisted breeding PHS resistance in wheat varieties.

In addition, four QTLs, *Qdor.hzau-2A*, *Qdor.hzau-4D*, *Qdor.hzau-5B* and *Qdor.hzau-5D*, were detected in only one of the four environments, on chromosomes 2A, 4D, 5B and 5D in this study. None of the four unstable QTLs was found in previous studies on seed dormancy or PHS resistance [6], indicating they might be novel QTLs for seed dormancy. Interestingly, preliminary results showed that the QTL *Qdor.hzau-5D* was also detected for heading date, flowering date and tillering angle in this DH population and co-localized with the vernalization-related gene *VRN1-5D* [41], indicating that this QTL might have pleiotropic effects. However, more evidences need to be explored in future research to further validate the authenticity of these QTLs.

## 4. Materials and Methods

### 4.1. Plant Materials and Trial Environments

The parents Yangmai16 (YM16) and Zhongmai895 (ZM895) and 174 DH lines were planted at Luoyang (Henan Province) in the 2020–2021 cropping season and at Wuhan (Hubei Province), Xiangyang (Hubei Province) and Luoyang (Henan Province) in the 2021–2022 cropping season, named LY21, WH22, XY22 and LY22 in this study, respectively. Field experiments were arranged in randomized complete blocks with three replications and each plot was 1 m single row in which 20 seeds were sown. The field management followed local agricultural practice. A panel of 412 wheat accessions was utilized for validation of the KASP marker for *Qdor.hzau-3D.2*, including 305 wheat cultivars and 107 wheat landraces collected from both China and other countries, representing global genetic diversity.

### 4.2. Phenotype Assessment

In this study, germination index (GI) and germination percentage (GP) were measured for seed dormancy [42,43]. In physiologically mature stage (hard dough stage), five spikes were harvested from different plants of each of the DH lines. The harvested spikes were air-dried for 2 days at room temperature, hand-threshed to avoid damage to the embryos of seeds, then stored at −20 °C to maintain dormancy until phenotyping [20]. Seeds were sterilized with 1% (*v/v*) of sodium hypochlorite for 10 min, followed by three rinses with sterile water. Fifty clean seeds were incubated in a 90 mm Petri dish with a filter paper and 8 mL of distilled water in the dark conditions for 7 days at 22 °C with three replications. Germinated seeds were counted daily and removed. GI and GP were calculated based on the following formula: GI = [(*n*1 × 7 + *n*2 × 6 + *n*3 × 5 + *n*4 × 4 + *n*5 × 3 + *n*6 × 2 + *n*7 × 1)/(*N* × 7)] × 100%; GP = [(*n*1 + *n*2 + *n*3 + *n*4 + *n*5 + *n*6 + *n*7)/*N*] × 100%. The parameters *n*1, *n*2, *n*3, *n*4, *n*5, *n*6 and *n*7 represent the number of germinated seed on day 1, day 2, day 3, day 4, day 5, day 6 and day 7, respectively. *N* represents the total of seeds.

For the wheat natural population, the spike germination percentage was obtained using the whole spike germination method, according to the procedure in the determination of pre-harvest sprouting in wheat (https://www.sdtdata.com/fx/fmrule/tsLibCard.doView, accessed on 15 August 2022). Briefly, five physiologically matured spikes were harvested and soaked in water for 4 h, then spikes were sterilized with 0.1% (*v/v*) of sodium hypochlorite for 5 min, followed by three rinses with sterile water. Spikes were incubated at 22 °C and RH100% (relative humidity) for 4 days, then dried at 80 °C for 2 days. The processed spikes were hand-threshed and the geminated seeds were counted. SGP was calculated by the formula: SGP = (germinated seeds)/(total seeds) × 100%.

Seeds harvested from the environment LY21 and XY22 were used for grain color assessment. The grain color was assessed according to the procedure provided by Imtiaz et al. [44]. About 30–40 seeds were soaked with 10 mL of 5% sodium hydroxide solution for 4 h at room temperature. Red grain cultivars showed dark-red color while white grains exhibited straw-yellow color. The number 0 and 1 were used to represent white and red DH lines for data analysis in this study, respectively.

### 4.3. Genotyping and Linkage Mapping

The parents and 174 DH lines were genotyped with the wheat 660K SNP array, and the genetic linkage map was constructed in the previous study [45]. There were 14,480 markers on this genetic map covered with 21 wheat chromosomes; the full length of the genetic map was 3681.7 cM and the average marker interval was 0.25 cM.

Markers for grain color loci *TaMyb10-A*, *TaMyb10-B* and *TaMyb10-D* were used for genotyping the parent YM16 and ZM895. In this study, markers for *TaMyb10-A* contained three pairs of primers, named *TaMyb10-A1-F/R* (F: 5′-CTACCAGCTCGTTTGGGAAG-3′, R: 5′-CTACCAGCTCGTTTGGGAAG-3), *TaMyb10-A2-F/R* (F: 5′-TTTCAATCGAGTGGGCATAA-3′, R: 5′-CCTGACGATGAGCTCCTCTT-3′) and *TaMyb10-A3-F/R* (F: 5′-TGTTATCACATGCTGATCCTGA-3′, R: 5′-TCCCTACATGGGAGACAGAGA-3′), respectively; markers for *TaMyb10-B* and *TaMyb10-D* were named *TaMyb10-B-F/R* (F: 5′-AGGAACCTGCAGTCTCACGG-3′, R: 5′-CTCGTGAACCCCCTCTGCT-3′) and *TaMyb10-D-F/R* (F: 5′-TAGGCCAACACCTTCTAAACG-3′, R: 5′-AGGCACACCAGCTTATTTGG-3′), respectively; primers information for these markers was obtained from previous studies [34,43].

### 4.4. Statistical Analysis

GI and GP correlation coefficients and analysis of variance (ANOVA) were obtained using QTL IciMapping version 4.2 [46]. The best linear unbiased estimation (BLUE) values were calculated by R package “lme4” using R4.2.2 software [47]; the mixed linear model was used for BLUE value calculation, the DH lines were used as fixed factors while the year, location and interaction between replication and location were used as random factors. The broad-sense heritability (*H^2^*) was estimated by the following formula: *H^2^* = Vg/(*Vg* + *Vge/e* + *Ve/re*), where *Vg*, *Vge* and *Ve* represent genotypic variance, the variance of genotype by environment and error variance, respectively; r and e represent the number of replications and environments, respectively.

### 4.5. QTL Analysis

GI and GP values in single environment and BLUE values across all environments for the 174 DH lines were used for QTL mapping. For grain color QTL detection, phenotypic data was used from environment LY21 and XY22. Inclusive composite interval mapping (ICIM-ADD) method under the bi-parental populations model was used for QTL analysis. QTLs with LOD score greater than 2.5 were reported and named using standard nomenclature for wheat. The physical positions of QTLs in this study were obtained based on the flanking markers using Chinese Spring v1.0 genome as reference from the Wheat Omics 1.0 website (http://202.194.139.32/, accessed on 20 October 2022). In this study, QTLs with large shared position or peaks of adjacent QTL distanced to less than 1 cM or overlapped with physical interval were regarded as the same QTL [48].

### 4.6. KASP Marker Development and Validation

The flanking marker *AX-108883716* linked to the QTL *Qdor.hzau-3D.2* was converted to a KASP assay. The KASP primers in this study were designed using the Primer Server tool of Wheat Omics 1.0 website. The forward primers contain FAM-primer and HEX-primer, they were 5′-GAAGGTGACCAAGTTCATGCTgtgagtgtactgaacttggttgT-3′ and 5′-GAAGGTCGGAGTCAACGGATTgtgagtgtactgaacttggttgC-3′, respectively; the reverse primer was 5′-GAATTCGGTGTCCAGGACCT-3′. The PCR reactions were performed in a total volume of 5.25 μL, containing 2.5 μL of KASP master mix (JasonGen, Beijing, China), 0.25 μL of primer mix (0.05 μL FAM primer, 0.05 μL HEX primer and 0.15 μL common reverse primer), 0.5 μL of genomic DNA (concentration of 100 ng/μL) and 2 μL of ddH_2_O. The PCR was performed on a BioRad CFX96 qPCR machine according to the following steps: 95 °C for 10 min, 10 touchdown cycles for 65–55 °C (decreasing by −1 °C per cycle), followed by 35 additional cycles (95 °C for 15 s and 55 °C for 1 min), and then read fluorescence data at 30 °C for 30 s. The data was downloaded from the inbuilt BioRad CFX96 software and analyzed with Microsoft Excel 2016.

## Figures and Tables

**Figure 1 plants-12-00759-f001:**
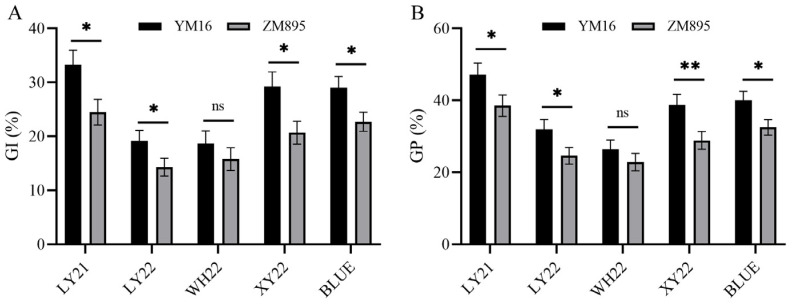
Effect of *Qdor.hzau-3D.2* on GI (**A**) and GP (**B**) in the YM16/ZM895 double haploid population. YM16 and ZM895 represented alleles of *Qdor.hzau-3D.2* from parent Yangmai16 and Zhongmai895, respectively; * *p* < 0.05; ** *p* < 0.01; ns, nonsignificant.

**Figure 2 plants-12-00759-f002:**
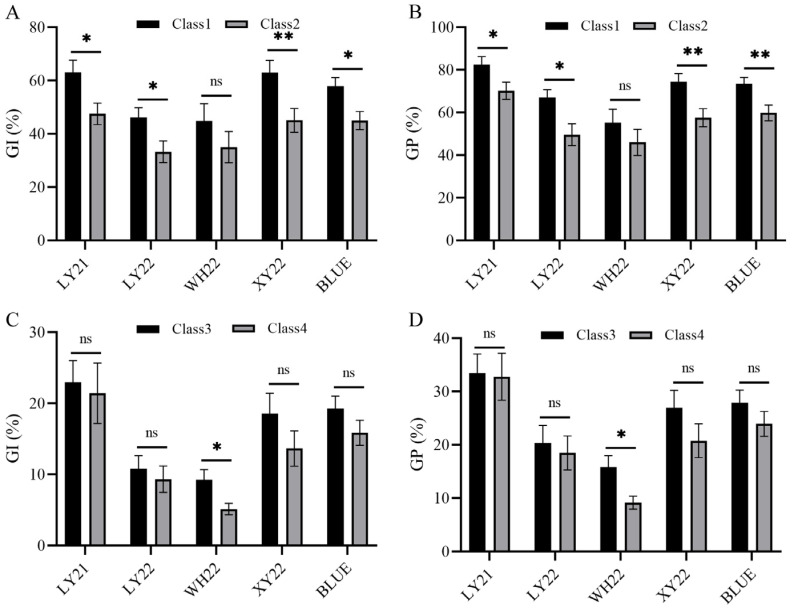
Effect of *Qdor.hzau-3D.2* in the absence or presence of grain color QTLs on GI (**A**,**C**) and GP (**B**,**D**) for all environments. Class 1 and 2 represented without and with the loci *Qdor.hzau-3D.2* in the absence of grain color QTLs, respectively; Class 3 and 4 represented without and with the loci *Qdor.hzau-3D.2* in the presence of grain color QTLs, respectively; * *p* < 0.05; ** *p* < 0.01; ns, nonsignificant.

**Figure 3 plants-12-00759-f003:**
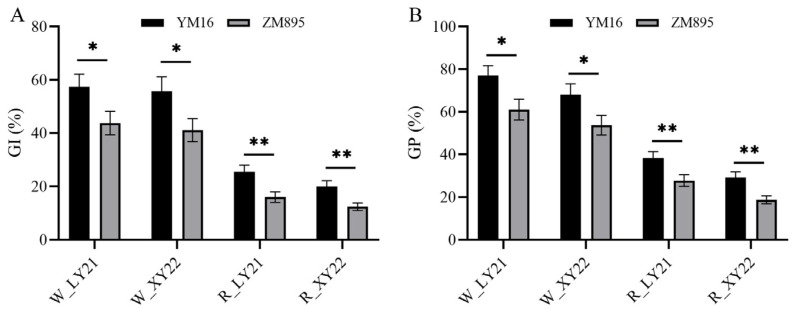
Effect of *Qdor.hzau-3D.2* on GI (**A**) and GP (**B**) in red and white-grained DH subpopulations. The uppercase letter W and R on the x-axis represented white- and red-grained DH subpopulations; YM16 and ZM895 represented alleles of *Qdor.hzau-3D.2* from Yangmai16 and Zhongmai895, respectively; * *p* < 0.05; ** *p* < 0.01.

**Figure 4 plants-12-00759-f004:**
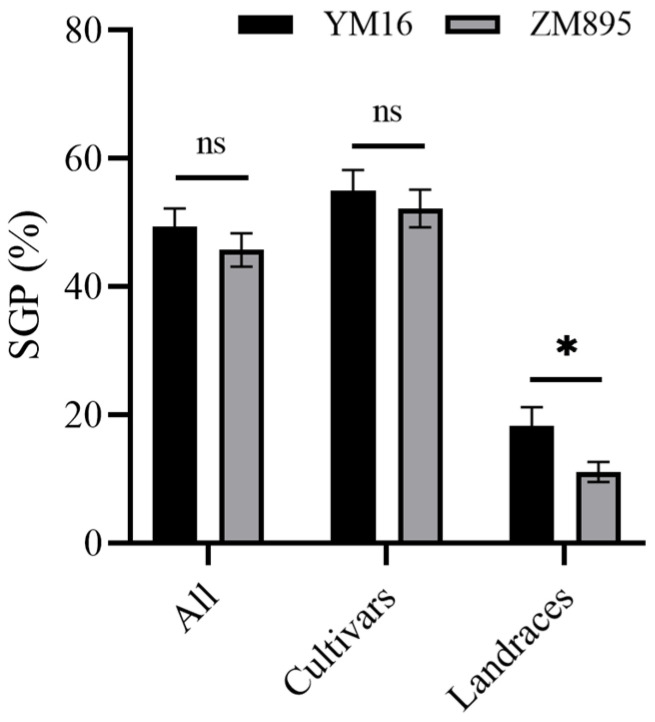
Effect of *Qdor.hzau-3D.2* on spike germination percentage in wheat natural population. YM16, wheat varieties carrying the C allele; ZM895, wheat varieties carrying the T allele; * *p* < 0.05; ns, nonsignificant.

**Table 1 plants-12-00759-t001:** GI of the YM16/ZM895 double haploid population and its parents in multiple environments.

Environment	Parent		DH Population			
	YM16 (%)	ZM895 (%)	Min (%)	Max (%)	Mean (%)	Heritability
LY21	7.52 ****	76.38	0.00	96.38	28.95	0.94
LY22	4.95 ****	78.67	0.10	74.00	16.83	0.78
WH22	3.11 ****	74.79	0.10	93.52	17.43	0.87
XY22	4.48 ****	78.48	0.29	92.57	25.02	0.84
BLUE	9.00 ****	80.00	5.54	79.12	25.90	0.88

LY21, Luoyang 2021; LY22, Luoyang 2022; WH22, Wuhan 2022; XY22, Xiangyang 2022; BLUE, the best linear unbiased estimation value across all environments. The same abbreviations were used in the following text in this study. **** *p* < 0.0001.

**Table 2 plants-12-00759-t002:** GP of the YM16/ZM895 double haploid population and its parents in multiple environments.

Environment	Parent		DH Population			
	YM16 (%)	ZM895 (%)	Min (%)	Max (%)	Mean (%)	Heritability
LY21	19.33 ****	88.67	0.00	98.67	43.12	0.84
LY22	9.33 ****	92.67	0.67	88.67	28.58	0.75
WH22	6.00 ****	80.44	0.67	95.33	25.07	0.84
XY22	9.33 ****	87.33	1.33	97.33	34.07	0.79
BLUE	14.95 ****	90.23	7.49	88.02	36.59	0.88

The environments were described in Table 1. **** *p* < 0.0001.

**Table 3 plants-12-00759-t003:** QTLs for GI and GP detected by ICIM in the YM16/ZM895 DH population.

QTL	Trait	Env	Position	Left Marker	Right Marker	LOD	PVE (%)	Add
*Qdor.hzau-2A*	GI	LY21	57	*AX-109313988*	*AX-109575090*	3.5	6.12	−0.06
*Qdor.hzau-3A*	GI	LY21	176	*AX-94633769*	*AX-109554358*	3.61	6.72	−0.06
		LY22	182	*AX-94455457*	*AX-111235128*	7.08	11.67	−0.06
		XY22	185	*AX-95119177*	*AX-111670577*	6.2	8.89	−0.07
		BLUE	185	*AX-95119177*	*AX-111670577*	8.54	11.78	−0.06
	GP	LY21	176	*AX-94633769*	*AX-109554358*	3.56	6.46	−0.07
		LY22	182	*AX-94455457*	*AX-111235128*	6.2	9.63	−0.08
		WH22	185	*AX-95119177*	*AX-111670577*	3.57	4.67	−0.05
		XY22	185	*AX-95119177*	*AX-111670577*	5.98	8.47	−0.08
		BLUE	185	*AX-95119177*	*AX-111670577*	7.44	10.01	−0.07
*Qdor.hzau-3D.1*	GI	LY21	176	*AX-110967455*	*AX-109649069*	8.15	15.31	−0.09
		LY22	180	*AX-110194478*	*AX-110515593*	13.67	23.27	−0.08
		WH22	182	*AX-94543851*	*AX-94850967*	12.71	28.3	−0.1
		XY22	176	*AX-110967455*	*AX-109649069*	14.03	21.99	−0.11
		BLUE	176	*AX-110967455*	*AX-109649069*	17.93	28.14	−0.09
	GP	LY21	180	*AX-110194478*	*AX-110515593*	11.05	21.08	−0.13
		LY22	180	*AX-110194478*	*AX-110515593*	13.93	22.95	−0.12
		WH22	182	*AX-94543851*	*AX-94850967*	19.37	31.94	−0.13
		XY22	176	*AX-110967455*	*AX-109649069*	13.22	20.01	−0.12
		BLUE	180	*AX-110194478*	*AX-110515593*	18.49	28.97	−0.12
*Qdor.hzau-3D.2*	GI	LY21	229	*AX-110937331*	*AX-108883716*	3.9	7.24	0.06
		LY22	227	*AX-110937331*	*AX-108883716*	3.83	6.59	0.04
		XY22	206	*AX-94926374*	*AX-110536745*	6.38	9.49	0.07
		BLUE	230	*AX-110937331*	*AX-108883716*	5.7	7.49	0.05
	GP	LY21	228	*AX-110937331*	*AX-108883716*	3.84	7.32	0.08
		LY22	226	*AX-110937331*	*AX-108883716*	3.35	5.6	0.06
		WH22	228	*AX-110937331*	*AX-108883716*	3.4	4.98	0.05
		XY22	206	*AX-94926374*	*AX-110536745*	6.97	10.34	0.09
		BLUE	230	*AX-110937331*	*AX-108883716*	7.36	9.94	0.07
*Qdor.hzau-4D*	GP	LY22	43	*AX-108763556*	*AX-110984743*	3.26	4.75	0.05
*Qdor.hzau-5B*	GP	LY21	124	*AX-94615114*	*AX-94962457*	2.58	4.04	0.06
*Qdor.hzau-5D*	GI	WH22	170	*AX-110586161*	*AX-108740226*	4.68	9.26	0.06
	GP	WH22	170	*AX-110586161*	*AX-108740226*	5.98	8.13	0.07

QTL, quantitative trait loci; Env, environment; Position, the genetic distance shown in cM; LOD, logarithm of odds; PVE, phenotypic variation explained by QTL; Add, additive effect with positive and negative values indicated an increasing effect from ZM895 and YM16, respectively.

**Table 4 plants-12-00759-t004:** QTLs for grain color detected by ICIM in the YM16/ZM895 DH population.

QTL	Env	Position	Left Marker	Right Marker	LOD	PVE (%)	Add
*Qgc.hzau-3A*	LY21	185	*AX-95119177*	*AX-111670577*	9.76	19.25	0.17
	XY22	185	*AX-95119177*	*AX-111670577*	10.35	21.11	0.19
*Qgc.hzau-3D*	LY21	182	*AX-94543851*	*AX-94850967*	14.99	30.44	0.23
	XY22	180	*AX-110194478*	*AX-110515593*	14.72	30.25	0.23

The same abbreviations were described in Table 3.

**Table 5 plants-12-00759-t005:** GI and GP of four genotypic combinations in the YM16/ZM895 DH population.

Type	Num	GI (%)					GP (%)				
		LY21	LY22	WH22	XY22	BLUE	LY21	LY22	WH22	XY22	BLUE
Class1	16	63.00	46.12	44.77	62.95	57.96	82.00	67.00	55.25	74.00	73.49
Class2	20	47.51	33.42	35.08	45.19	45.12	70.00	49.58	45.96	58.00	59.99
Class3	35	23.01	10.76	9.33	18.57	19.24	34.00	20.32	15.87	24.00	28.00
Class4	23	21.46	9.28	5.02	13.71	15.82	33.00	18.58	9.09	21.00	24.12

Class 1, 2, 3 and 4 were described in Figure 2; Num, number of DH lines for class 1, 2, 3 and 4, respectively.

## Data Availability

Not applicable.

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
