# Peer review of "QTL Mapping for Wheat Seed Dormancy in a Yangmai16/Zhongmai895 Double Haploid Population"

_plants, 2023, doi:10.3390/plants12040759_

Round 1

Reviewer 1 Report

QTL mapping for wheat seed dormancy in a Yangmai16/Zhongmai895 Double Haploid populations  is very important to accelerate breeding for improving grain yield and quality in wheat. However, the paper suffers for some limits: (1) Please explain why you have used Wuhan, Xiangyang and Luoyang for the field experiment. In my opinion, these experiment sites were underrepresentation for wheat production in China. (2) The parents and 174 DH were used in the previous study and the novelty in this study is limited.

Author Response

 (1) Please explain why you have used Wuhan, Xiangyang and Luoyang for the field experiment. In my opinion, these experiment sites were underrepresentation for wheat production in China.

Response: Thanks for the comments. As reviewer said, these three sites might be underrepresentation for wheat production in China. However, for seed dormancy and preharvest-sprouting trait, we think they are representative sites as they have contrast whether condition during harvest season. Wuhan is hot and frequently to have rain when the wheat is maturing and Xiangyang has less rain during harvest season. For Luoyang, this site is fit for wheat production as they have lower temperature and much less rain compared to Wuhan. In addition, we actually had planed more other sites but we could not harvest the seeds because of region block for COVID-19. Nonetheless, we think these three sites could be used for our experiment.

(2) The parents and 174 DH were used in the previous study and the novelty in this study is limited. 

Response: As reviewer said, the parents and DH lines have been used in previous study. However, after careful investigation, we could not find publication about the seed dormancy or preharvest-sprouting using the same population. It would be better to publish the result with a population first time but we think it would be reasonable to use the same population to dissect the genetics of different trait. In addition, we found a novel locus for seed dormancy which also could increase the novelty of this study. We are trying to get reviewer's agreement and further comments are most welcome.

Reviewer 2 Report

Seed dormancy will affect the pre-harvest sprouting (PHS) in wheat, which is an important trait for grain yield and quality. The authors detected QTLs for seed dormancy and grain color using a doubled haploid (DH) population. A novel stable QTL Qdor.hzau-3D.2 for seed dormancy was identified and validated in this study. The results provide the clues for the improvement for marker-assisted wheat breeding for PHS resistance in the future. Several comments as below.

1. If possible, the candidate genes of QTL Qdor.hzau-3D.2 could be analyzed and discussed. It will give more information to readers.

2. The significant difference test between parents of YM16 and ZM895 should be added in Table 1 and Table 2.

3. The authors showed that the effect of Qdor.hzau-3D.2 would be expressed only in the absence of Qdor.hzau-3A/Qgc.hzau-3A and Qdor.hzau-3D.1/Qgc.hzau-3D while it would be suppressed when the grain color related QTLs were present. The results are interesting. Does Qdor.hzau-3D.2 link to Qdor.hzau-3D.1/Qgc.hzau-3D? Please give more discussion.

4. The formula of GI is incorrect in Methods. Please check it.

Author Response

  1. If possible, the candidate genes of QTL Qdor.hzau-3D.2 could be analyzed and discussed. It will give more information to readers.

Response: Thanks for the comment and it’s really helpful. We have added the candidate genes information. See lines 117-120.

  1. The significant difference test between parents of YM16 and ZM895 should be added in Table 1 and Table 2.

Response: We have finished the test and made changes in the Table1 and Table2.

  1. The authors showed that the effect of hzau-3D.2 would be expressed only in the absence of Qdor.hzau-3A/Qgc.hzau-3A and Qdor.hzau-3D.1/Qgc.hzau-3D while it would be suppressed when the grain color related QTLs were present. The results are interesting. Does Qdor.hzau-3D.2 link to Qdor.hzau-3D.1/Qgc.hzau-3D? Please give more discussion.

Response: Thanks for the nice comments. We are also interested with this result. To further investigate the genetic interaction between the two QTLs, near isogenic lines will be developed and also used for fine-mapping the causal gene of Qdor.hzau-3D.2. Please also see the changes in lines 220-224 in text.

  1. The formula of GI is incorrect in Methods. Please check it.

   Response: We have double checked the formula we used and compared it with others in references, and we found we have used the same formula as others. However, we might have missed several spaces to separate the number. Now we have made changes in the text. Thanks for the comments.

Reviewer 3 Report

The work deals with seed dormancy - an important aspect of cereal physiology of great importance for plant breeding and agriculture. The pre-harvest sprouting is a polygenic trait highly influenced by environment therefore studying the factors determining resistance to PHS, and the creation of markers that can help track the trait in the breeding material are important research topics. Although the work did not lead to the detection of a major loci differentiating cultivars, it describes the research reliably and systematizes the knowledge. I see this manuscript as a important contribution to the global PHS discussion.

However, I do have two comments that I would like to address:

11. Your results are based on the observations in 4 environments. Actually what I see is 3 environments Luoyang, Wuhan, Xiangyang  and 2 cropping seasons (Luoyang ) or 1 cropping season (Wuhan, Xiangyang) . Could you comment on this and include respective weather descriptions for each region. This could be useful in checking if over-matured seeds in harvesting are the result of climatic conditions in LY21.

2.       Why the spikes were not subjected to humid and warm conditions before the GI and GR were scored? Do you think that the conditions used in the experiment were adequate for phenotypic development and evaluation?

Author Response

  1. Your results are based on the observations in 4 environments. Actually what I see is 3 environments Luoyang, Wuhan, Xiangyang and 2 cropping seasons (Luoyang ) or 1 cropping season (Wuhan, Xiangyang) . Could you comment on this and include respective weather descriptions for each region. This could be useful in checking if over-matured seeds in harvesting are the result of climatic conditions in LY21.

Response: Thanks for the comments. We defined different environments as location and cropping season combinations, that is, 2 cropping seasons for Luoyang are treated as two different environments conditions. The weather conditions could partly explain the over-matured seeds in LY21 as high temperature with dry air condition was presented during harvesting time in 2021, Luoyang. Unfortunately, we could not get the weather condition in detail as private. In addition, it might be attributed to several days delay because of COVID19 region block during harvesting time.

  1. Why the spikes were not subjected to humid and warm conditions before the GI and GR were scored? Do you think that the conditions used in the experiment were adequate for phenotypic development and evaluation?

Response: The reviewer may want to express the method for preharvest-sprouting trait evaluation. In this case, spikes will be immersed into water and keep humid and warm conditions. However, for seed dormancy, we used mature seed threshed from the spikes for GI/GR evaluation, and this is also widely used by others research. We hope we understand the reviewer comment correctly. If not, we are happy to give more explanation at our best.

Reviewer 4 Report

The manuscript "QTL mapping for wheat seed dormancy in a Yangmai16/Zhongmai895 Double Haploid population" describes the mapping of seven QTLs for seed dormancy on chromosomes 2A, 3A, 3D, 4D, 5B and 5D over four environments, and also two QTLs on chromosomes 3A and 3D for grain color using a doubled haploid (DH) population. Among these QTLs, Qdor.hzau-3D.2 has never been reported before and probably a novel locus from the weakly dormant parent ZM895, and then a KASP marker linked with Qdor.hzau-3D.2 was developed and tested in a wheat natural population. In my opinion, it was an interesting manuscript with good information and could be accepted for publication after some minor revisions.

Minor corrections

1.     Line 39, changed “on the third and fourth chromosome homologous groups” to “on chromosomes of the third and fourth chromosome homologous groups”

2.     Lines 41 and 43, since genes TaPHS1 and TaMKK3 have been cloned, they should not be “the candidate gene”, could be “the causal gene”

3.     Line 69-71, it was not clear to me that a KASP marker for which QTL? Clarify in the sentence;

4.     Lines 109-111, 308-309, please clarify which wheat reference genome? And which version?  “According to the flanking marker and physical position,” should be in the plural

5.     Lines 124, 168, “P” should be Italic

6.     Lines 138-139, changed “on 3A and 3D” to “on chromosomes 3A and 3D”; delete the word “for”

7.  Lines 188, 196, 261, not clear to me for the words “(Cosgrove, #1580)

8.  Line 233, delete “(unpublished results)

Lines 249-250, “five spikes were harvested from each DH lines”, five spikes were from the same plant or different plants?

Author Response

  1. Line 39, changed “on the third and fourth chromosome homologous groups” to “on chromosomes of the third and fourth chromosome homologous groups”

Response: We have changed as suggested. See line 39.

  1. Lines 41 and 43, since genes TaPHS1 and TaMKK3 have been cloned, they should not be “the candidate gene”, could be “the causal gene”

Response: We have changed as suggested. See lines 41 and 43.

  1. Line 69-71, it was not clear to me that a KASP marker for which QTL? Clarify in the sentence;

Response: We have made change to clarify this. See line 71.

  1. Lines 109-111, 308-309, please clarify which wheat reference genome? And which version?  “According to the flanking marker and physical position,” should be in the plural

Response: We have made changes to clarify this. See lines 110 and 317.

  1. Lines 124, 168, “P” should be Italic

Response: We have made changes as suggested. See lines 128 and 172, and two more positions of P Italic in lines 186 and 202.

  1. Lines 138-139, changed “on 3A and 3D” to “on chromosomes 3A and 3D”; delete the word “for”

Response: We have made changes and deleted the word as suggested. See lines 142-143.

  1. Lines 188, 196, 261, not clear to me for the words “(Cosgrove, #1580)”

Response: We are sorry to make this error because of reference manager Endnote. Now it’s fixed. See lines 191, 200, and 269.

  1. Line 233, delete “(unpublished results)”

Response: As suggested, we deleted the words. See line 241.

Lines 249-250, “five spikes were harvested from each DH lines”, five spikes were from the same plant or different plants?

Response: We clarify this as “five spikes were harvested from different spikes of each DH lines”. See line 258 in text.